# Network Pruning via Transformable Architecture Search

**Xuanyi Dong**[†‡*]**, Yi Yang**[†]
[†]ReLER, CAI, University of Technology Sydney, [‡]Baidu Research
xuanyi.dong@student.uts.edu.au; yi.yang@uts.edu.au

## Abstract

Network pruning reduces the computation costs of an over-parameterized network without performance damage. Prevailing pruning algorithms pre-define the width and depth of the pruned networks, and then transfer parameters from the unpruned network to pruned networks. To break the structure limitation of the pruned networks, we propose to apply neural architecture search to search directly for a network with flexible channel and layer sizes. The number of the channels/layers is learned by minimizing the loss of the pruned networks. The feature map of the pruned network is an aggregation of K feature map fragments (generated by K networks of different sizes), which are sampled based on the probability distribution. The loss can be back-propagated not only to the network weights, but also to the parameterized distribution to explicitly tune the size of the channels/layers. Specifically, we apply channel-wise interpolation to keep the feature map with different channel sizes aligned in the aggregation procedure. The maximum probability for the size in each distribution serves as the width and depth of the pruned network, whose parameters are learned by knowledge transfer, e.g., knowledge distillation, from the original networks. Experiments on CIFAR-10, CIFAR-100 and ImageNet demonstrate the effectiveness of our new perspective of network pruning compared to traditional network pruning algorithms. Various searching and knowledge transfer approaches are conducted to show the effectiveness of the two components. Code is at: `https://github.com/D-X-Y/NAS-Projects`.

## 1   Introduction

Deep convolutional neural networks (CNNs) have become wider and deeper to achieve high performance on different applications [17, 22, 48]. Despite their great success, it is impractical to deploy them to resource constrained devices, such as mobile devices and drones. A straightforward solution to address this problem is using network pruning [29, 12, 13, 20, 18] to reduce the computation cost of over-parameterized CNNs. A typical pipeline for network pruning, as indicated in Fig. 1(a), is achieved by removing the redundant filters and then fine-tuning the slashed networks, based on the original networks. Different criteria for the importance of the filters are applied, such as L2-norm of the filter [30], reconstruction error [20], and learnable scaling factor [32]. Lastly, researchers apply various fine-tuning

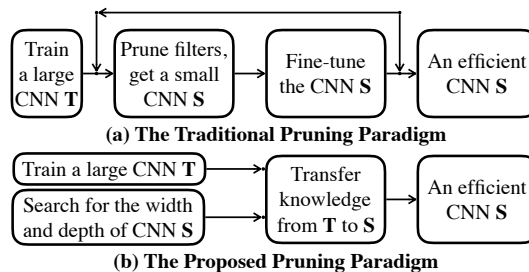

**(a) The Traditional Pruning Paradigm**

**(b) The Proposed Pruning Paradigm**

Figure 1: A comparison between the typical pruning paradigm and the proposed paradigm.

---

[*]This work was done when Xuanyi Dong was a research intern at Baidu Research.

strategies [30, 18] for the pruned network to efficiently transfer the parameters of the unpruned networks and maximize the performance of the pruned networks.

Traditional network pruning approaches achieve effective impacts on network compacting while maintaining accuracy. Their network structure is intuitively designed, e.g., pruning 30% filters in each layer [30, 18], predicting sparsity ratio [15] or leveraging regularization [2]. The accuracy of the pruned network is upper bounded by the hand-crafted structures or rules for structures. To break this limitation, we apply Neural Architecture Search (NAS) to turn the design of the architecture structure into a learning procedure and propose a new paradigm for network pruning as explained in Fig. 1(b).

Prevailing NAS methods [31, 48, 8, 4, 40] optimize the network topology, while the focus of this paper is automated network size. In order to satisfy the requirements and make a fair comparison between the previous pruning strategies, we propose a new NAS scheme termed Transformable Architecture Search (TAS). TAS aims to search for the best size of a network instead of topology, regularized by minimization of the computation cost, e.g., floating point operations (FLOPs). The parameters of the searched/pruned networks are then learned by knowledge transfer [21, 44, 46].

TAS is a differentiable searching algorithm, which can search for the width and depth of the networks effectively and efficiently. Specifically, different candidates of channels/layers are attached with a learnable probability. The probability distribution is learned by back-propagating the loss generated by the pruned networks, whose feature map is an aggregation of K feature map fragments (outputs of networks in different sizes) sampled based on the probability distribution. These feature maps of different channel sizes are aggregated with the help of channel-wise interpolation. The maximum probability for the size in each distribution serves as the width and depth of the pruned network.

In experiments, we show that the searched architecture with parameters transferred by knowledge distillation (KD) outperforms previous state-of-the-art pruning methods on CIFAR-10, CIFAR-100 and ImageNet. We also test different knowledge transfer approaches on architectures generated by traditional hand-crafted pruning approaches [30, 18] and random architecture search approach [31]. Consistent improvements on different architectures demonstrate the generality of knowledge transfer.

## 2   Related Studies

Network pruning [29, 33] is an effective technique to compress and accelerate CNNs, and thus allows us to deploy efficient networks on hardware devices with limited storage and computation resources. A variety of techniques have been proposed, such as low-rank decomposition [47], weight pruning [14, 29, 13, 12], channel pruning [18, 33], dynamic computation [9, 7] and quantization [23, 1]. They lie in two modalities: unstructured pruning [29, 9, 7, 12] and structured pruning [30, 20, 18, 33].

***Unstructured*** pruning methods [29, 9, 7, 12] usually enforce the convolutional weights [29, 14] or feature maps [7, 9] to be sparse. The pioneers of unstructured pruning, LeCun et al. [29] and Hassibi et al. [14], investigated the use of the second-derivative information to prune weights of shallow CNNs. After deep network was born in 2012 [28], Han et al. [12, 13, 11] proposed a series of works to obtain highly compressed deep CNNs based on L2 regularization. After this development, many researchers explored different regularization techniques to improve the sparsity while preserve the accuracy, such as L0 regularization [35] and output sensitivity [41]. Since these unstructured methods make a big network sparse instead of changing the whole structure of the network, they need dedicated design for dependencies [11] and specific hardware to speedup the inference procedure.

***Structured*** pruning methods [30, 20, 18, 33] target the pruning of convolutional filters or whole layers, and thus the pruned networks can be easily developed and applied. Early works in this field [2, 42] leveraged a group Lasso to enable structured sparsity of deep networks. After that, Li et al. [30] proposed the typical three-stage pruning paradigm (training a large network, pruning, re-training). These pruning algorithms regard filters with a small norm as unimportant and tend to prune them, but this assumption does not hold in deep nonlinear networks [43]. Therefore, many researchers focus on better criterion for the informative filters. For example, Liu et al. [32] leveraged a L1 regularization; Ye et al. [43] applied a ISTA penalty; and He et al. [19] utilized a geometric median-based criterion. In contrast to previous pruning pipelines, our approach allows the number of channels/layers to be explicitly optimized so that the learned structure has high-performance and low-cost.

Besides the criteria for informative filters, the importance of network structure was suggested in [33]. Some methods implicitly find a data-specific architecture [42, 2, 15], by automatically determining

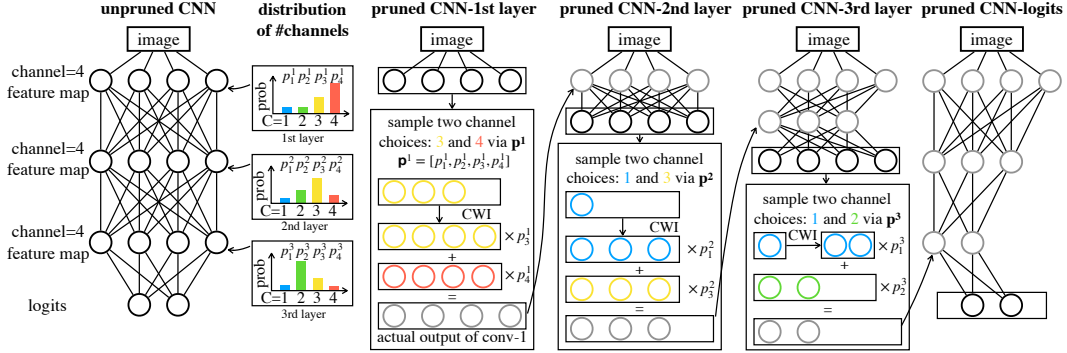

Figure 2: Searching for the width of a pruned CNN from an unpruned three-layer CNN. Each convolutional layer is equipped with a learnable distribution for the size of the channels in this layer, indicated by $\boldsymbol{p}^i$ on the left side. The feature map for every layer is built sequentially by the layers, as shown on the right side. For a specific layer, K (2 in this example) feature maps of different sizes are sampled according to corresponding distribution and combined by channel-wise interpolation (CWI) and weighted sum. This aggregated feature map is fed as input to the next layer.

the pruning and compression ratio of each layer. In contrast, we explicitly discover the architecture using NAS. Most previous NAS algorithms [48, 8, 31, 40] automatically discover the topology structure of a neural network, while we focus on searching for the depth and width of a neural network. Reinforcement learning (RL)-based [48, 3] methods or evolutionary algorithm-based [40] methods are possible to search networks with flexible width and depth, however, they require huge computational resources and cannot be directly used on large-scale target datasets. Differentiable methods [8, 31, 4] dramatically decrease the computation costs but they usually assume that the number of channels in different searching candidates is the same. TAS is a differentiable NAS method, which is able to efficiently search for a transformable networks with flexible width and depth.

Network transformation [5, 10, 3] also studied the depth and width of networks. Chen et al. [5] manually widen and deepen a network, and proposed Net2Net to initialize the lager network. Ariel et al. [10] proposed a heuristic strategy to find a suitable width of networks by alternating between shrinking and expanding. Cai et al. [3] utilized a RL agent to grow the depth and width of CNNs, while our TAS is a differentiable approach and can not only enlarge but also shrink CNNs.

Knowledge transfer has been proven to be effective in the literature of pruning. The parameters of the networks can be transferred from the pre-trained initialization [30, 18]. Minnehan et al. [37] transferred the knowledge of uncompressed network via a block-wise reconstruction loss. In this paper, we apply a simple KD approach [21] to perform knowledge transfer, which achieves robust performance for the searched architectures.

## 3   Methodology

Our pruning approach consists of three steps: (1) training the unpruned large network by a standard classification training procedure. (2) searching for the depth and width of a small network via the proposed TAS. (3) transferring the knowledge from the unpruned large network to the searched small network by a simple KD approach [21]. We will introduce the background, show the details of TAS, and explain the knowledge transfer procedure.

### 3.1   Transformable Architecture Search

Network channel pruning aims to reduce the number of channels in each layer of a network. Given an input image, a network takes it as input and produces the probability over each target class. Suppose $\boldsymbol{X}$ and $\boldsymbol{O}$ are the input and output feature tensors of the $l$-th convolutional layer (we take 3-by-3 convolution as an example), this layer calculates the following procedure:

$$\boldsymbol{O}_j = \sum\nolimits_{k=1}^{c_{in}} \boldsymbol{X}_{k,:,:} * \boldsymbol{W}_{j,k,:,:} \quad \text{where } 1 \le j \le c_{out}, \tag{1}$$

where $\boldsymbol{W} \in \mathbb{R}^{c_{out} \times c_{in} \times 3 \times 3}$ indicates the convolutional kernel weight, $c_{in}$ is the input channel, and $c_{out}$ is the output channel. $\boldsymbol{W}_{j,k,:,:}$ corresponds to the $k$-th input channel and $j$-th output channel. $*$ denotes the convolutional operation. Channel pruning methods could reduce the number of $c_{out}$, and consequently, the $c_{in}$ in the next layer is also reduced.

**Search for width**. We use parameters $\alpha \in \mathbb{R}^{|\mathbb{C}|}$ to indicate the distribution of the possible number of channels in one layer, indicated by $\mathbb{C}$ and $\max(\mathbb{C}) \leq c_{out}$. The probability of choosing the $j$-th candidate for the number of channels can be formulated as:

$$p_j = \frac{\exp(\alpha_j)}{\sum_{k=1}^{|\mathbb{C}|} \exp(\alpha_k)} \quad \text{where} \ \ 1 \leq j \leq |\mathbb{C}|, \tag{2}$$

However, the sampling operation in the above procedure is non-differentiable, which prevents us from back-propagating gradients through $p_j$ to $\alpha_j$. Motivated by [8], we apply Gumbel-Softmax [26, 36] to soften the sampling procedure to optimize $\alpha$:

$$\hat{p}_j = \frac{\exp((\log(p_j) + \boldsymbol{o}_j)/\tau)}{\sum_{k=1}^{|\mathbb{C}|} \exp((\log(p_k) + \boldsymbol{o}_k)/\tau)} \quad \text{s.t.} \ \ \boldsymbol{o}_j = -\log(-\log(u)) \ \& \ u \sim \mathcal{U}(0,1), \tag{3}$$

where $\mathcal{U}(0,1)$ means the uniform distribution between 0 and 1. $\tau$ is the softmax temperature. When $\tau \to 0$, $\hat{p} = [\hat{p}_1, ..., \hat{p}_j, ...]$ becomes one-hot, and the Gumbel-softmax distribution drawn from $\hat{p}$ becomes identical to the categorical distribution. When $\tau \to \infty$, the Gumbel-softmax distribution becomes a uniform distribution over $\mathbb{C}$. The feature map in our method is defined as the weighted sum of the original feature map fragments with different sizes, where weights are $\hat{p}$. Feature maps with different sizes are aligned by channel wise interpolation (CWI) so as for the operation of weighted sum. To reduce the memory costs, we select a small subset with indexes $\mathbb{I} \subseteq [|\mathbb{C}|]$ for aggregation instead of using all candidates. Additionally, the weights are re-normalized based on the probability of the selected sizes, which is formulated as:

$$\hat{\boldsymbol{O}} = \sum_{j \in \mathbb{I}} \frac{\exp((\log(p_j) + \boldsymbol{o}_j)/\tau)}{\sum_{k \in \mathbb{I}} \exp((\log(p_k) + \boldsymbol{o}_k)/\tau)} \times \text{CWI}(\boldsymbol{O}_{1:\mathbb{C}_j,:,:}, \max(\mathbb{C}_{\mathbb{I}})) \ \text{s.t.} \ \mathbb{I} \sim \mathcal{T}_{\hat{p}}, \tag{4}$$

where $\mathcal{T}_{\hat{p}}$ indicates the multinomial probability distribution parameterized by $\hat{p}$. The proposed CWI is a general operation to align feature maps with different sizes. It can be implemented in many ways, such a 3D variant of spatial transformer network [25] or adaptive pooling operation [16]. In this paper, we choose the 3D adaptive average pooling operation [16] as CWI[2] , because it brings no extra parameters and negligible extra costs. We use Batch Normalization [24] before CWI to normalize different fragments. Fig. 2 illustrates the above procedure by taking $|\mathbb{I}| = 2$ as an example.

*Discussion w.r.t. the sampling strategy in Eq.* (4). This strategy aims to largely reduce the memory cost and training time to an acceptable amount by only back-propagating gradients of the sampled architectures instead of all architectures. Compared to sampling via a uniform distribution, the applied sampling method (sampling based on probability) could weaken the gradients difference caused by per-iteration sampling after multiple iterations.

**Search for depth**. We use parameters $\beta \in \mathbb{R}^L$ to indicate the distribution of the possible number of layers in a network with $L$ convolutional layers. We utilize a similar strategy to sample the number of layers following Eq. (3) and allow $\beta$ to be differentiable as that of $\alpha$, using the sampling distribution $\hat{q}_l$ for the depth $l$. We then calculate the final output feature of the pruned networks as an aggregation from all possible depths, which can be formulated as:

$$\boldsymbol{O}_{out} = \sum_{l=1}^{L} \hat{q}_l \times \text{CWI}(\hat{\boldsymbol{O}}^l, C_{out}), \tag{5}$$

where $\hat{\boldsymbol{O}}^l$ indicates the output feature map via Eq. (4) at the $l$-th layer. $C_{out}$ indicates the maximum sampled channel among all $\hat{\boldsymbol{O}}^l$. The final output feature map $\boldsymbol{O}_{out}$ is fed into the last classification layer to make predictions. In this way, we can back-propagate gradients to both width parameters $\alpha$ and depth parameters $\beta$.

**Searching objectives**. The final architecture $\mathcal{A}$ is derived by selecting the candidate with the maximum probability, learned by the architecture parameters $\mathbb{A}$, consisting of $\alpha$ for each layers and $\beta$. The goal of our TAS is to find an architecture $\mathcal{A}$ with the minimum validation loss $\mathcal{L}_{val}$ after trained by minimizing the training loss $\mathcal{L}_{train}$ as:

$$\min_{\mathcal{A}} \mathcal{L}_{val}(\omega_{\mathcal{A}}^*, \mathcal{A}) \quad \text{s.t.} \quad \omega_{\mathcal{A}}^* = \arg\min_{\omega} \mathcal{L}_{train}(\omega, \mathcal{A}), \tag{6}$$

where $\omega_{\mathcal{A}}^*$ indicates the optimized weights of $\mathcal{A}$. The training loss is the cross-entropy classification loss of the networks. Prevailing NAS methods [31, 48, 8, 4, 40] optimize $\mathcal{A}$ over network candidates with different typologies, while our TAS searches over candidates with the same typology structure as well as smaller width and depth. As a result, the validation loss in our search procedure includes not only the classification validation loss but also the penalty for the computation cost:

$$\mathcal{L}_{val} = -\log\left(\frac{\exp(\boldsymbol{z}_y)}{\sum_{j=1}^{|\boldsymbol{z}|} \exp(\boldsymbol{z}_j)}\right) + \lambda_{cost}\mathcal{L}_{cost}, \tag{7}$$

where $\boldsymbol{z}$ is a vector denoting the output logits from the pruned networks, $y$ indicates the ground truth class of a corresponding input, and $\lambda_{cost}$ is the weight of $\mathcal{L}_{cost}$. The cost loss encourages the computation cost of the network (e.g., FLOP) to converge to a target $R$ so that the cost can be dynamically adjusted by setting different $R$. We used a piece-wise computation cost loss as:

$$\mathcal{L}_{cost} = \begin{cases} \log(\mathbb{E}_{cost}(\mathbb{A})) & F_{cost}(\mathbb{A}) > (1+t) \times R \\ 0 & (1-t) \times R < F_{cost}(\mathbb{A}) < (1+t) \times R \\ -\log(\mathbb{E}_{cost}(\mathbb{A})) & F_{cost}(\mathbb{A}) < (1-t) \times R \end{cases}, \tag{8}$$

where $\mathbb{E}_{cost}(\mathbb{A})$ computes the expectation of the computation cost, based on the architecture parameters $\mathbb{A}$. Specifically, it is the weighted sum of computation costs for all candidate networks, where the weight is the sampling probability. $F_{cost}(\mathbb{A})$ indicates the actual cost of the searched architecture, whose width and depth are derived from $\mathbb{A}$. $t \in [0, 1]$ denotes a toleration ratio, which slows down the speed of changing the searched architecture. Note that we use FLOP to evaluate the computation cost of a network, and it is readily to replace FLOP with other metric, such as latency [4].

We show the overall algorithm in Alg. 1. During searching, we forward the network using Eq. (5) to make both weights and architecture parameters differentiable. We alternatively minimize $\mathcal{L}_{train}$ on the training set to optimize the pruned networks' weights and $\mathcal{L}_{val}$ on the validation set to optimize the architecture parameters $\mathbb{A}$. After searching, we pick up the number of channels with the maximum probability as width and the number of layers with the maximum probability as depth. The final searched network is constructed by the selected width and depth. This network will be optimized via KD, and we will introduced the details in Sec. 3.2.

---

**Algorithm 1** The TAS Procedure

---

**Input:** split the training set into two disjoint sets: $\mathbb{D}_{train}$ and $\mathbb{D}_{val}$
1: **while** not converge **do**
2:     Sample batch data $\mathbb{D}_t$ from $\mathbb{D}_{train}$
3:     Calculate $\mathcal{L}_{train}$ on $\mathbb{D}_t$ to update network weights
4:     Sample batch data $\mathbb{D}_v$ from $\mathbb{D}_{val}$
5:     Calculate $\mathcal{L}_{val}$ on $\mathbb{D}_v$ via Eq. (7) to update $\mathbb{A}$
6: **end while**
7: Derive the searched network from $\mathbb{A}$
8: Randomly initialize the searched network and optimize it by KD via Eq. (10) on the training set

---

## 3.2 Knowledge Transfer

Knowledge transfer is important to learn a robust pruned network, and we employ a simple KD algorithm [21] on a searched network architecture. This algorithm encourages the predictions $\boldsymbol{z}$ of the small network to match soft targets from the unpruned network via the following objective:

$$\mathcal{L}_{\text{match}} = -\sum_{i=1}^{|\boldsymbol{z}|} \frac{\exp(\hat{\boldsymbol{z}}_i/T)}{\sum_{j=1}^{|\boldsymbol{z}|} \exp(\hat{\boldsymbol{z}}_j/T)} \log\left(\frac{\exp(\boldsymbol{z}_i/T)}{\sum_{j=1}^{|\boldsymbol{z}|} \exp(\boldsymbol{z}_j/T)}\right), \tag{9}$$

where $T$ is a temperature, and $\hat{\boldsymbol{z}}$ indicates the logit output vector from the pre-trained unpruned network. Additionally, it uses a softmax with cross-entropy loss to encourage the small network to predict the true targets. The final objective of KD is as follows:

$$\mathcal{L}_{\text{KD}} = -\lambda \log\left(\frac{\exp(\boldsymbol{z}_y)}{\sum_{j=1}^{|\boldsymbol{z}|} \exp(\boldsymbol{z}_j)}\right) + (1-\lambda)\mathcal{L}_{\text{match}} \quad \text{s.t.} \ 0 \leq \lambda \leq 1, \tag{10}$$

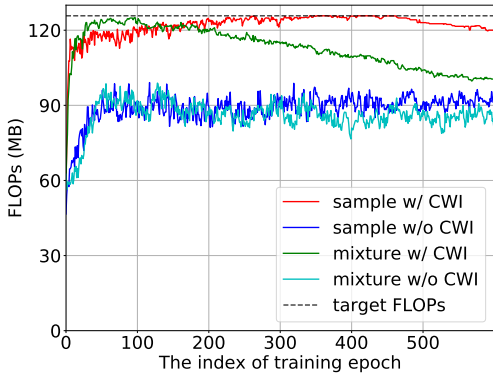

(a) The FLOPs of the searched network over epochs when we do not constrain the FLOPs ($\lambda_{cost} = 0$).

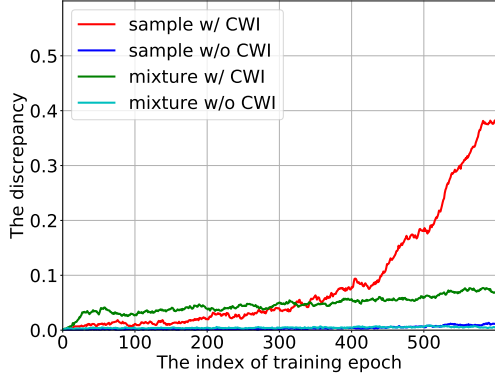

(b) The mean discrepancy over epochs when we do not constrain the FLOPs ($\lambda_{cost} = 0$).

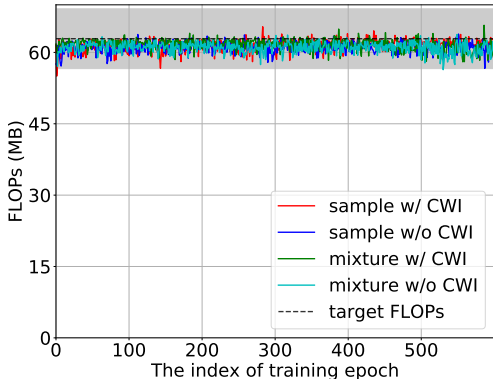

(c) The FLOPs of the searched network over epochs when we constrain the FLOPs ($\lambda_{cost} = 2$).

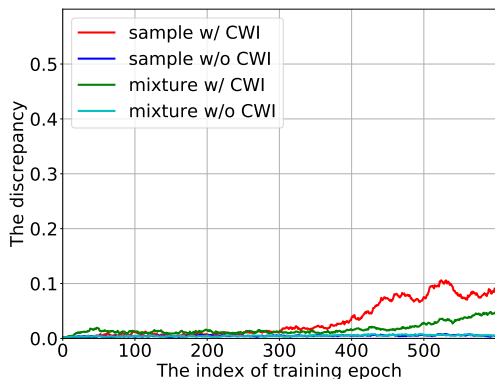

(d) The mean discrepancy over epochs when we constrain the FLOPs ($\lambda_{cost} = 2$).

Figure 3: The impact of different choices to make architecture parameters differentiable.

where $y$ indicates the true target class of a corresponding input. $\lambda$ is the weight of loss to balance the standard classification loss and soft matching loss. After we obtain the searched network (Sec. 3.1), we first pre-train the unpruned network and then optimize the searched network by transferring from the unpruned network via Eq. (10).

# 4 Experimental Analysis

We introduce the experimental setup in Sec. 4.1. We evaluate different aspects of TAS in Sec. 4.2, such as hyper-parameters, sampling strategies, different transferring methods, etc. Lastly, we compare TAS with other state-of-the-art pruning methods in Sec. 4.3.

## 4.1 Datasets and Settings

**Datasets.** We evaluate our approach on CIFAR-10, CIFAR-100 [27] and ImageNet [6]. CIFAR-10 contains 50K training images and 10K test images with 10 classes. CIFAR-100 is similar to CIFAR-10 but has 100 classes. ImageNet contains 1.28 million training images and 50K test images with 1000 classes. We use the typical data augmentation of these three datasets. On CIFAR-10 and CIFAR-100, we randomly crop 32×32 patch with 4 pixels padding on each border, and we also apply the random horizontal flipping. On ImageNet, we use the typical random resized crop, randomly changing the brightness / contrast / saturation, and randomly horizontal flipping for data augmentation. During evaluation, we resize the image into 256×256 and center crop a 224×224 patch.

**The search setting.** We search the number of channels over {0.3, 0.4, 0.5, 0.6, 0.7, 0.8, 0.9, 1.0} of the original number in the unpruned network. We search the depth within each convolutional stage.

We sample $|\mathbb{I}| = 2$ candidates in Eq. (4) to reduce the GPU memory cost during searching. We set $R$ according to the FLOPs of the compared pruning algorithms and set $\lambda_{cost}$ of 2. We optimize the weights via SGD and the architecture parameters via Adam. For the weights, we start the learning rate from 0.1 and reduce it by the cosine scheduler [34]. For the architecture parameters, we use the constant learning rate of 0.001 and a weight decay of 0.001. On both CIFAR-10 and CIFAR-100, we train the model for 600 epochs with the batch size of 256. On ImageNet, we train ResNets [17] for 120 epochs with the batch size of 256. The toleration ratio $t$ is always set as 5%. The $\tau$ in Eq. (3) is linearly decayed from 10 to 0.1.

Table 1: The accuracy on CIFAR-100 when pruning about 40% FLOPs of ResNet-32.

|  | FLOPs | accuracy |
|---|---|---|
| Pre-defined | 41.1 MB | 68.18 % |
| Pre-defined w/ Init | 41.1 MB | 69.34 % |
| Pre-defined w/ KD | 41.1 MB | 71.40 % |
| Random Search | 42.9 MB | 68.57 % |
| Random Search w/ Init | 42.9 MB | 69.14 % |
| Random Search w/ KD | 42.9 MB | 71.71 % |
| TAS† | 42.5 MB | 68.95 % |
| TAS† w/ Init | 42.5 MB | 69.70 % |
| TAS† w/ KD (TAS) | 42.5 MB | **72.41 %** |

**Training.** For CIFAR experiments, we use SGD with a momentum of 0.9 and a weight decay of 0.0005. We train each model by 300 epochs, start the learning rate at 0.1, and reduce it by the cosine scheduler [34]. We use the batch size of 256 and 2 GPUs. When using KD on CIFAR, we use $\lambda$ of 0.9 and the temperature $T$ of 4 following [46]. For ResNet models on ImageNet, we follow most hyper-parameters as CIFAR, but use a weight decay of 0.0001. We use 4 GPUs to train the model by 120 epochs with the batch size of 256. When using KD on ImageNet, we set $\lambda$ as 0.5 and $T$ as 4 on ImageNet.

## 4.2 Case Studies

In this section, we evaluate different aspects of our proposed TAS. We also compare it with different searching algorithm and knowledge transfer method to demonstrate the effectiveness of TAS.

**The effect of different strategies to differentiate** $\alpha$. We apply our TAS on CIFAR-100 to prune ResNet-56. We try two different aggregation methods, i.e., using our proposed CWI to align feature maps or not. We also try two different kinds of aggregation weights, i.e., Gumbel-softmax sampling as Eq. (3) (denoted as "sample" in Fig. 3) and vanilla-softmax as Eq. (2) (denoted as "mixture" in Fig. 3). Therefore, there are four different strategies, i.e., with/without CWI combining with Gumbel-softmax/vanilla-softmax. Suppose we do not constrain the computational cost, then the architecture parameters should be optimized to find the maximum width and depth. This is because such network will have the maximum capacity and result in the best performance on CIFAR-100. We try all four strategies with and without using the constraint of computational cost. We show the results in Fig. 3c and Fig. 3a. When we do not constrain the FLOPs, our TAS can successfully find the best architecture should have a maximum width and depth. However, other three strategies failed. When we use the FLOP constraint, we can successfully constrain the computational cost in the target range. We also investigate discrepancy between the highest probability and the second highest probability in Fig. 3d and Fig. 3b. Theoretically, a higher discrepancy indicates that the model is more confident to select a certain width, while a lower discrepancy means that the model is confused and does not know which candidate to select. As shown in Fig. 3d, with the training procedure going, our TAS becomes more confident to select the suitable width. In contrast, strategies without CWI can not optimize the architecture parameters; and "mixture with CWI" shows a worse discrepancy than ours.

Table 2: Results of different configurations when prune ResNet-32 on CIFAR-10 with one V100 GPU. "#SC" indicates the number of selected channels. "H" indicates hours.

| #SC | Search Time | Memory | Train Time | FLOPs | Accuracy |
|---|---|---|---|---|---|
| $\|\mathbb{I}\|$=1 | 2.83 H | 1.5GB | 0.71 H | 23.59 MB | 89.85% |
| $\|\mathbb{I}\|$=2 | 3.83 H | 2.4GB | 0.84 H | 38.95 MB | 92.98% |
| $\|\mathbb{I}\|$=3 | 4.94 H | 3.4GB | 0.67 H | 39.04 MB | 92.63% |
| $\|\mathbb{I}\|$=5 | 7.18 H | 5.1GB | 0.60 H | 37.08 MB | 93.18% |
| $\|\mathbb{I}\|$=8 | 10.64 H | 7.3GB | 0.81 H | 38.28 MB | 92.65% |

**Comparison w.r.t. structure generated by different methods** in Table 1. "Pre-defined" means pruning a fixed ratio at each layer [30]. "Random Search" indicates an NAS baseline used in [31]. "TAS†" is our proposed differentiable searching algorithm. We make two observations: (1) searching can find a better structure using different knowledge transfer methods; (2) our TAS is superior to the NAS random baseline.

Table 3: Comparison of different pruning algorithms for ResNet on CIFAR. "Acc" = accuracy, "FLOPs" = FLOPs (pruning ratio), "TAS (D)" = searching for depth, "TAS (W)" = searching for width, "TAS" = searching for both width and depth.

| Depth | Method | CIFAR-10 | | | CIFAR-100 | | |
|---|---|---|---|---|---|---|---|
| | | Prune Acc | Acc Drop | FLOPs | Prune Acc | Acc Drop | FLOPs |
| 20 | LCCL [7] | 91.68% | 1.06% | 2.61E7 (36.0%) | 64.66% | 2.87% | 2.73E7 (33.1%) |
| | SFP [18] | 90.83% | 1.37% | 2.43E7 (42.2%) | 64.37% | 3.25% | 2.43E7 (42.2%) |
| | FPGM [19] | 91.09% | 1.11% | 2.43E7 (42.2%) | 66.86% | 0.76% | 2.43E7 (42.2%) |
| | TAS (D) | 90.97% | 1.91% | 2.19E7 (46.2%) | 64.81% | 3.88% | 2.19E7 (46.2%) |
| | TAS (W) | 92.31% | 0.57% | 1.99E7 (51.3%) | 68.08% | 0.61% | 1.92E7 (52.9%) |
| | TAS | 92.88% | 0.00% | 2.24E7 (45.0%) | 68.90% | -0.21% | 2.24E7 (45.0%) |
| 32 | LCCL [7] | 90.74% | 1.59% | 4.76E7 (31.2%) | 67.39% | 2.69% | 4.32E7 (37.5%) |
| | SFP [18] | 92.08% | 0.55% | 4.03E7 (41.5%) | 68.37% | 1.40% | 4.03E7 (41.5%) |
| | FPGM [19] | 92.31% | 0.32% | 4.03E7 (41.5%) | 68.52% | 1.25% | 4.03E7 (41.5%) |
| | TAS (D) | 91.48% | 2.41% | 4.08E7 (41.0%) | 66.94% | 3.66% | 4.08E7 (41.0%) |
| | TAS (W) | 92.92% | 0.96% | 3.78E7 (45.4%) | 71.74% | -1.12% | 3.80E7 (45.0%) |
| | TAS | 93.16% | 0.73% | 3.50E7 (49.4%) | 72.41% | -1.80% | 4.25E7 (38.5%) |
| 56 | PFEC [30] | 93.06% | -0.02% | 9.09E7 (27.6%) | — | — | — |
| | LCCL [7] | 92.81% | 1.54% | 7.81E7 (37.9%) | 68.37% | 2.96% | 7.63E7 (39.3%) |
| | AMC [15] | 91.90% | 0.90% | 6.29E7 (50.0%) | — | — | — |
| | SFP [18] | 93.35% | 0.56% | 5.94E7 (52.6%) | 68.79% | 2.61% | 5.94E7 (52.6%) |
| | FPGM [19] | 93.49% | 0.42% | 5.94E7 (52.6%) | 69.66% | 1.75% | 5.94E7 (52.6%) |
| | TAS | 93.69% | 0.77% | 5.95E7 (52.7%) | 72.25% | 0.93% | 6.12E7 (51.3%) |
| 110 | LCCL[7] | 93.44% | 0.19% | 1.66E8 (34.2%) | 70.78% | 2.01% | 1.73E8 (31.3%) |
| | PFEC [30] | 93.30% | 0.20% | 1.55E8 (38.6%) | — | — | — |
| | SFP [18] | 92.97% | 0.70% | 1.21E8 (52.3%) | 71.28% | 2.86% | 1.21E8 (52.3%) |
| | FPGM [19] | 93.85% | -0.17% | 1.21E8 (52.3%) | 72.55% | 1.59% | 1.21E8 (52.3%) |
| | TAS | 94.33% | 0.64% | 1.19E8 (53.0%) | 73.16% | 1.90% | 1.20E8 (52.6%) |
| 164 | LCCL[7] | 94.09% | 0.45% | 1.79E8 (27.40%) | 75.26% | 0.41% | 1.95E8 (21.3%) |
| | TAS | 94.00% | 1.47% | 1.78E8 (28.10%) | 77.76% | 0.53% | 1.71E8 (30.9%) |

**Comparison w.r.t. different knowledge transfer methods** in Table 1. The first line in each block does not use any knowledge transfer method. "w/ Init" indicates using pre-trained unpruned network as initialization. "w/ KD" indicates using KD. From Table 1, knowledge transfer methods can consistently improve the accuracy of pruned network, even if a simple method is applied (Init). Besides, KD is robust and improves the pruned network by more than 2% accuracy on CIFAR-100.

**Searching width vs. searching depth.** We try (1) only searching depth ("TAS (D)"), (2) only searching width ("TAS (W)"), and (3) searching both depth and width ("TAS") in Table 3. Results of only searching depth are worse than results of only searching width. If we jointly search for both depth and width, we can achieve better accuracy with similar FLOP than both searching depth and searching width only.

**The effect of selecting different numbers of architecture samples $\mathbb{I}$ in Eq.** (4). We compare different numbers of selected channels in Table 2 and did experiments on a single NVIDIA Tesla V100. The searching time and the GPU memory usage will increase linearly to $|\mathbb{I}|$. When $|\mathbb{I}|=1$, since the re-normalized probability in Eq. (4) becomes a constant scalar of 1, the gradients of parameters $\alpha$ will become 0 and the searching failed. When $|\mathbb{I}|>1$, the performance for different $|\mathbb{I}|$ is similar.

**The speedup gain.** As shown in Table 2, TAS can finish the searching procedure of ResNet-32 in about 3.8 hours on a single V100 GPU . If we use evolution strategy (ES) or random searching methods, we need to train network with many different candidate configurations one by one and then evaluate them to find a best. In this way, much more computational costs compared to our TAS are

Table 4: Comparison of different pruning algorithms for different ResNets on ImageNet.

| Model | Method | Top-1 Prune Acc | Top-1 Acc Drop | Top-5 Prune Acc | Top-5 Acc Drop | FLOPs | Prune Ratio |
|---|---|---|---|---|---|---|---|
| ResNet-18 | LCCL [7] | 66.33% | 3.65% | 86.94% | 2.29% | 1.19E9 | 34.6% |
| | SFP [18] | 67.10% | 3.18% | 87.78% | 1.85% | 1.06E9 | 41.8% |
| | FPGM [19] | 68.41% | 1.87% | 88.48% | 1.15% | 1.06E9 | 41.8% |
| | TAS | 69.15% | 1.50% | 89.19% | 0.68% | 1.21E9 | 33.3% |
| ResNet-50 | SFP [18] | 74.61% | 1.54% | 92.06% | 0.81% | 2.38E9 | 41.8% |
| | CP [20] | - | - | 90.80% | 1.40% | 2.04E9 | 50.0% |
| | Taylor [38] | 74.50% | 1.68% | - | - | 2.25E9 | 44.9% |
| | AutoSlim [45] | 76.00% | - | - | - | 3.00E9 | 26.6% |
| | FPGM [19] | 75.50% | 0.65% | 92.63% | 0.21% | 2.36E9 | 42.2% |
| | TAS | 76.20% | 1.26% | 93.07% | 0.48% | 2.31E9 | 43.5% |

required. A possible solution to accelerate ES or random searching methods is to share parameters of networks with different configurations [39, 45], which is beyond the scope of this paper.

### 4.3 Compared to the state-of-the-art

**Results on CIFAR** in Table 3. We prune different ResNets on both CIFAR-10 and CIFAR-100. Most previous algorithms perform poorly on CIFAR-100, while our TAS consistently outperforms then by more than 2% accuracy in most cases. On CIFAR-10, our TAS outperforms the state-of-the-art algorithms on ResNet-20,32,56,110. For example, TAS obtains 72.25% accuracy by pruning ResNet-56 on CIFAR-100, which is higher than 69.66% of FPGM [19]. For pruning ResNet-32 on CIFAR-100, we obtain greater accuracy and less FLOP than the unpruned network. We obtain a slightly worse performance than LCCL [7] on ResNet-164. It because there are $8^{163} \times 18^3$ candidate network structures to searching for pruning ResNet-164. It is challenging to search over such huge search space, and the very deep network has the over-fitting problem on CIFAR-10 [17].

**Results on ImageNet** in Table 4. We prune ResNet-18 and ResNet-50 on ImageNet. For ResNet-18, it takes about 59 hours to search for the pruned network on 4 NVIDIA Tesla V100 GPUs. The training time of unpruned ResNet-18 costs about 24 hours, and thus the searching time is acceptable. With more machines and optimized implementation, we can finish TAS with less time cost. We show competitive results compared to other state-of-the-art pruning algorithms. For example, TAS prunes ResNet-50 by 43.5% FLOPs, and the pruned network achieves 76.20% accuracy, which is higher than FPGM by 0.7. Similar improvements can be found when pruning ResNet-18. Note that we directly apply the hyper-parameters on CIFAR-10 to prune models on ImageNet, and thus TAS can potentially achieve a better result by carefully tuning parameters on ImageNet.

Our proposed TAS is a preliminary work for the new network pruning pipeline. This pipeline can be improved by designing more effective searching algorithm and knowledge transfer method. We hope that future work to explore these two components will yield powerful compact networks.

## 5 Conclusion

In this paper, we propose a new paradigm for network pruning, which consists of two components. For the first component, we propose to apply NAS to search for the best depth and width of a network. Since most previous NAS approaches focus on the network topology instead the network size, we name this new NAS scheme as Transformable Architecture Search (TAS). Furthermore, we propose a differentiable TAS approach to efficiently and effectively find the most suitable depth and width of a network. For the second component, we propose to optimize the searched network by transferring knowledge from the unpruned network. In this paper, we apply a simple KD algorithm to perform knowledge transfer, and conduct other transferring approaches to demonstrate the effectiveness of this component. Our results show that new efforts focusing on searching and transferring may lead to new breakthroughs in network pruning.

## Footnotes

[2]The formulation of the selected CWI: suppose $\boldsymbol{B} = \text{CWI}(\boldsymbol{A}, C_{out})$, where $\boldsymbol{B} \in \mathbb{R}^{C_{out}HW}$ and $\boldsymbol{A} \in \mathbb{R}^{CHW}$; then $\boldsymbol{B}_{i,h,w} = \text{mean}(\boldsymbol{A}_{s:e-1,h,w})$, where $s = \lfloor \frac{i \times C}{C_{out}} \rfloor$ and $e = \lceil \frac{(i+1) \times C}{C_{out}} \rceil$. We tried other forms of CWI, e.g., bilinear and trilinear interpolation. They obtain similar accuracy but are much slower than our choice.

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
