[Reviews · NeurIPS 2019]

Reviewer 1



Originality: Using a differentiable architecture search for pruning a network is new to me and it makes sense. Also using interpolation for fusing different channel sizes makes sense and it can possibly be used for other applications too. Quality: The contribution of the paper seems clear and the proposed methodology makes sense. Experimental results show that the proposed pruning can lead to a good trade-off between computational cost and accuracy. Clarity: The presentation of the paper should be improved. For instance channel wise interpolation is abbreviated sometimes with (CWI) and other times with (CHI), making the reading very confusing. Also, the actual way that CHI is performed is not very clear. It can be intuited by Fig. 2 but a proper formulation is missing. Significance: In my opinion the contribution of this paper is important because it show that is possible to cast network pruning in terms of differentiable architecture search and results place the method among the most promising pruning approaches. Additional comments: - From tab.1, the use of knowledge distillation (KD) seems to be important for good results. However, KD can be used to improve any pruning approach. Thus in this sense it is not clear if the good performance of the proposed method are due to the network architecture optimization or the knowledge distillation. If it's the second case, then the contributions of the paper will be reduced. - Fig.2 helped me to fully understand the contribution. However, there should be a clearer formulation of the approach too. - In l.50 the authors talk about optimizing the number of channels but do not talk about the number of layers. This is a bit confusing because it is not clear what is the final aim of the paper. This kind of problems can be found in several points in the paper. It seems like multiple people with different understanding wrote different parts of the paper. - Authors should compare with other approaches also in terms of training time. This method seems computationally quite intense during training. - From my understanding, in this work channels are grouped together based just on their order. This means that there can be other combination of channels that can outperform those predefined. One could argue that this constraint is maintained during training and it can induce to group channels in a meaning way. However, this is not true in case of starting the optimisation from a pre-trained network as it seem to be the case. Final decision: I read other reviews and rebuttal. Some of the answers to my questions are not fully satisfactory: - Q2.1: I saw the comparison with and without KD in figure 1, however, I wanted to see the influence of this factor in the final results. That is, the proposed method without KD would still be superior to others? The authors show results for a specif configuration in tab.2 and it seems that for the other methods using KD produces worse results. Is there any reason for that or it is a typo? - Q2.2: In this question I wanted to see more convincing results than a single experiment. However authors did not include any new experiment. Globally I still consider the paper in a positive way, however, I would like to see those two points clarified in a final version.

Reviewer 2



1) the paper motivates that the current network pruning approaches try the same architectural design whereas a more efficient approach could be found with new architecture. 2) paper organization was good and easy to follow 3) the approach is a bit non-intuitive: example why combine the different feature maps at all? why only depth and width search? It took some time to understand but the solution seems to be out-of-the-box like thinking. 4) the results are solid i.e. less FLOPs and better accuracy on imagenet compared to the previous approaches on network pruning

Reviewer 3



The writing of the article could be improved. Besides a few typographical errors across the paper (lines 72, 160, 160), there are some other parts where sentences could be rephrased, for example: 24: 'apply them' -> 'deploy them' 47: rephrase whole sentence 68: 'to develop the powerful networks'-> 'to deploy the deep networks' 87: Do not use 'while' as 'though' unless it is at the beginning of a sentence. 98: Rephrase whole sentence. The style of the article could also be improved. State more clearly the main contributions of the paper in the introductory sections and use Fig. 1a and Fig. 1b, instead of saying 'the first line of Fig. 1'. ---------------------------------------------------------------------------------------------- The proposed method is quite original, as if attempting to 'grow' a reduced network, instead of pruning a larger one. The NAS method is also very elegant mathematically; since it is setup as a differentiable problem, now the error can be propagated and gradient descend family of methods can be used to search a locally optimal solution (a structure in this case) in an efficient way. Having said that, yet this is another case of apply relaxation to a hard problem and call it a day. In fact, it is not easy to spot how much of the proposed TAS method is new or just a variant, a minor incremental improvement, or an use case, of the work already presented at [1]. [1] Dong, Xuanyi, and Yi Yang. "Searching for a robust neural architecture in four gpu hours." Proceedings of the IEEE Conference on Computer Vision and Pattern Recognition. 2019.

Reviewer 4



EDIT: I have read the other reviews and the rebuttal. Thanks to the authors for clarifying my questions. I am happy to raise my score, and urge the authors to make sure the clarity concerns are addressed in the draft. Clarity: I struggled with parts of the draft regarding clarity. Some specific points that I may have misunderstood: - I could not find a clear definition of CHI. Please define this clearly in the draft. If it has many possible interpretations, then specify the goal. Is CHI == CWI? If so, please explain this in a bit more detail as well. An example may be helpful. - How was the cost of the network estimated? In particular was is the difference between E_cost(A) and F(A) and how is this made into a differentiable loss? Please specify. - The role of distillation could have been made clearer. In particular, the authors could specify whether the new network inherits (1) weights, (2) architectures from the networks trained earlier in the TAS procedure. - The train / test / validation splits need to be clarified in the experimental section. This is especially important, since TAS trains on validation data. Quality: There are serious concerns regarding the quality of the experiments. - TAS uses validation data to optimize the architecture of the networks. Even though the validation data is not used directly to optimize the weights, it can still have a very significant influence on the learned weights via the bilevel optimization scheme. Therefore, it is correct to see the validation data as part of the training set for the purposes of evaluation of overfitting. I could not tell from the current draft whether the reported results are evaluated on the test or training sets. Moreoever, given that the validation set was not specified (from what I could tell!), it is hard to be sure whether the reported numbers reflect overfitting or not. - In the experiment on the effect of strategies to differentiate alpha, in was unclear why the choice was made to not constrain the computation cost. This seems like the least informative choice, and the fact that the chosen method is the only one that succeeds in this context doesn't make it obvious why it should succeed when you aim to constrain the computational cost. The authors could make this more convincing by including experiments with constrained costs. Significance: Given the concerns with the evaluation, it is hard to assess the significance of the work. Even so, the improvements were modest over competitor methods and raise concerns about the impact of the methods moving forward. Originality: The method is original, although not completely distinct from previous works, see Louizos et al. 2018. This would not be a major concern, if the experimental results were more interpretable and robust.

[Author Response · NeurIPS 2019]

We thank all four reviewers' time, effort, and valuable suggestions. We will (I) add the suggested experiments and
comparisons, (II) explain more intuition behind various design, (III) do our best to proofread our paper in revision.
────────────────────────**To Reviewer #2 (R2)**────────────────────────
**Q2.1: Compare results with/without KD.** We have compared TAS and other basic baselines with/without KD in Table 1 in
our paper, which shows KD can improve about 2% accuracy when pruning ResNet-32 on CIFAR-100. Additionally, we
compare advanced methods: SFP[15] 68.4% (with) VS 69.1% (without); FPGM[16] 68.5% (with) VS 69.3% (without).
**Q2.2: Is the applied sampling strategy the best?** The sampling procedure aims to largely reduce the memory cost and
training time to an acceptable amount by only back-propagating gradients of the sampled architectures instead of all
architectures. Compared to sampling via a uniform distribution, the applied sampling method (sampling based on
probability) can weaken the gradients difference caused by per-iteration sampling after multiple iterations.
**Q2.3: Effect of selecting different numbers of architecture samples $|\mathbb{I}|$.** As suggested, we compare different numbers of
selected channels in the table on the right. The searching time and the

Table 1: Results of different configurations when prune
ResNet-32 on CIFAR-10 with one V100 GPU. "#SC" indi-
cates the number of selected channels. "H" indicates hours.

| #SC | Search Time | Memory | Train Time | FLOPs | Accuracy |
|---|---|---|---|---|---|
| $|\mathbb{I}|$=1 | 2.83 H | 1.5GB | 0.71 H | 23.59 MB | 89.85% |
| $|\mathbb{I}|$=2 | 3.83 H | 2.4GB | 0.84 H | 38.95 MB | 92.98% |
| $|\mathbb{I}|$=3 | 4.94 H | 3.4GB | 0.67 H | 39.04 MB | 92.63% |
| $|\mathbb{I}|$=5 | 7.18 H | 5.1GB | 0.60 H | 37.08 MB | 93.18% |
| $|\mathbb{I}|$=8 | 10.64 H | 7.3GB | 0.81 H | 38.28 MB | 92.65% |

GPU memory usage will increase linearly to $|\mathbb{I}|$. When $|\mathbb{I}|$=1, since the
re-normalized probability in Eq.(4) becomes a constant scalar of 1, the
gradients of parameters $\alpha$ will become 0 and the searching failed. When
$|\mathbb{I}|$>1, the performance for different $|\mathbb{I}|$ is similar.
**Q2.4: Does different channel-wise interpolation (CWI) affect the perfor-**
**mance? and its formulation.** As pointed in L-140∼L-142, the proposed
CWI is a general operation to align feature maps with different sizes. We
use adaptive average pool (AAP) [13] in our experiments, and its formulation is $\widehat{\mathbf{O}}_{i,h,w}$=mean($\mathbf{O}_{s:e-1,h,w}$), where
$s = \lfloor \frac{i \times C}{C_{out}} \rfloor$ and $e = \lceil \frac{(i+1) \times C}{C_{out}} \rceil$. $\mathbf{O} \in \mathbb{R}^{CHW}$ and $\widehat{\mathbf{O}} \in \mathbb{R}^{C_{out}HW}$ are input and output tensors of CWI. We did try other
forms of CWI, such as bilinear and trilinear interpolation methods. They can obtain similar performance but are much
slower than our choice. It is interesting to analyze other CWI operations, which will be explored in future work.
────────────────────────**To Reviewer #3 (R3)**────────────────────────
**Q3.1: Why only search depth and width?** Researches on network architecture mainly focus on two aspects, i.e., network
topology and network size. Most NAS methods automated the design of network topology, which outperforms manually
designed topology. Most pruning methods manually set the network size. However, there is not an efficient way to
automate the tuning procedure of network size. In our paper, we target on automating the network size design, which is
challenging and helps to further boost performance. We are the first to search for the network size in a differentiable way.
**Q3.2: What is the intuition behind CWI?** We apply CWI to align feature map fragments with different channel sizes
to the *same* channel size. In this way, we can combine the fragments together and optimize the sampling probability
for each architecture. Please see Q2.4 for the specific case of CWI used in our experiments.
**Q3.3: What is the intuition of $|\mathbb{I}|$=2?** As analyzed in Q2.3 and L-204 in the paper, we choose $|\mathbb{I}|$=2 because it costs
the minimum searching time and GPU memory usage to find suitable width and depth for a network.
────────────────────────**To Reviewer #4 (R4)**────────────────────────
Many thanks for your valuable comments. We have carefully proofread the paper following your suggestion.
**Q4.1: Difference with [6](CVPR'19).** Even though [6] and our paper both use similar technique to differentiate the
sampling procedure by Gumbel Softmax, we target on different problems: our TAS focuses on network pruning by
searching width and depth, while [6] focuses on searching CNN topology in a general way.
**Q4.2: Compare with EA/RL with speedup gain.** Given the short period of rebuttal, we can only implement co-variance
matrix adaptation evolution strategy (CMA-ES) to optimize parameters $\alpha$, which represents the width and depth
selection. The estimated searching cost is over 1000 GPU hours. We would add the results once the CMA-ES algorithm
finished. In sum, our TAS is much faster than CMA-ES (3.8H VS 1000H) and Random Search (3.8H VS 20H).
────────────────────────**To Reviewer #5 (R5)**────────────────────────
**Q5.1: Clarify CWI.** 'CHI' is a typo and will be replaced by 'CWI'. Please find its formulation in Q2.4.
**Q5.2: Clarify the cost.** As indicated in L172, we use FLOPs to represent one network cost. $F(\mathbb{A})$ indicates FLOPs
of a network, whose width and depth are derived from $\arg\max(\mathbb{A})$ (explained in L179∼181). $\mathbb{E}_{cost}(\mathbb{A})$ is the weighted
sum of FLOPs for all candidate networks, where the weight is the sampling probability of the corresponding candidate.
**Q5.3: Clarify the distillation.** The unpruned network is trained on the whole training set. The new searched network
is derived from variable $\mathbb{A}$, which selects the best width/depth as L179∼L182. This network is trained from scratch
using the whole training set, with KD loss to distill knowledge from the pre-trained unpruned network.
**Q5.4: Clarification on validation.** During searching, we split 50% of training set as $\mathbb{D}_{train}$ (the training set in Alg.1)
and the rest 50% training set as $\mathbb{D}_{val}$ (the validation set in Alg.1) following the standard NAS setting [3,6,28]. After
obtaining the searched network (Step 7 in Alg.1), we randomly initialize this searched network and train it using
$\mathbb{D}_{train} \cup \mathbb{D}_{val}$. The reported accuracy is evaluated on the test set, which is never used during searching or training.
**Q5.5: Why not constrain FLOP in Fig.3?** As suggested, we did 4 experiments in Fig.3 with the computation cost
constraint $\mathcal{L}_{cost}$ to prune about 50% FLOPs of ResNet-110. We observe that with $\mathcal{L}_{cost}$, different cases have similar
relative performances but lower computation costs. Given the limited space, we can not include the new figure in
rebuttal. These results and more ablation/case studies suggested in Q2.3/Q4.2 will be added in the next revision.

[Meta-Review · NeurIPS 2019]

Reviewers are not entirely satisfied with your response, however, they are leaning to a positive overall opinion,. Hence I think your paper can be accepted provided (and I am really trusting on you, as there is no way to obligue you) you commit to address in as much as possible the open issues raised by reviewers.